

# The role and proteomic analysis of ethylene in hydrogen gas-induced adventitious rooting development in cucumber (*Cucumis sativus* L.) explants

Dengjing Huang[1], Biting Bian[1], Meiling Zhang[2], Chunlei Wang[1], Changxia Li[1] and Weibiao Liao[1]

[1] College of Horticulture, Gansu Agricultural University, Lanzhou, PR China, Lanzhou, China
[2] College of Science, Gansu Agricultural University, Lanzhou, China

## ABSTRACT

Previous studies have shown that both hydrogen gas ($H_2$) and ethylene (ETH) play positive roles in plant adventitious rooting. However, the relationship between $H_2$ and ETH during this process has not been explored and remains insufficiently understood. In this study, cucumber (*Cucumis sativus* L.) was used to explore the proteomic changes in ETH-$H_2$-induced rooting. Our results show that hydrogen-rich water (HRW) and ethylene-releasing compound (ethephon) at proper concentrations promote adventitious rooting, with maximal biological responses occurring at 50% HRW or 0.5 μM ethephon. ETH inhibitors aminoethoxyvinylglycine (AVG) and $AgNO_3$ cause partial inhibition of adventitious rooting induced by $H_2$, suggesting that ETH might be involved in $H_2$-induced adventitious rooting. According to two-dimensional electrophoresis (2-DE) and mass spectrometric analyses, compared with the control, 9 proteins were up-regulated while 15 proteins were down-regulated in HRW treatment; four proteins were up-regulated while 10 proteins were down-regulated in ethephon treatment; and one protein was up-regulated while nine proteins were down-regulated in HRW+AVG treatment. Six of these differentially accumulated proteins were further analyzed, including photosynthesis -related proteins (ribulose-1,5-bisphosphate carall boxylase smsubunit (Rubisco), sedoheptulose-1,7-bisphosphatase (SBPase), oxygen-evolving enhancer protein (OEE1)), amino and metabolism-related protein (threonine dehydratase (TDH)), stress response-related protein (cytosolic ascorbate peroxidase (CAPX)), and folding, modification and degradation-related protein (protein disulfide-isomerase (PDI)). Moreover, the results of real-time PCR about the mRNA levels of these genes in various treatments were consistent with the 2-DE results. Therefore, ETH may be the downstream signaling molecule during $H_2$- induced adventitious rooting and proteins Rubisco, SBPase, OEE1, TDH, CAPX and PDI may play important roles during the process.

Corresponding author
Weibiao Liao, liaowb@gsau.edu.cn

## INTRODUCTION

Adventitious roots are formed from non-root tissue such as leaves, stems and hypocotyls. Adventitious roots usually increase the area of roots, heightening their abilities to absorb nutrients and support plants. Additionally, adventitious roots play an important role in promoting the vegetative propagation of fine varieties and the agriculture and forestry industries. Thus, exploring the mechanism of adventitious root formation is both necessary and meaningful. The roles of signal molecules in adventitious root development have recently become a hot topic of research. Some signal molecules including auxin (*Bai et al., 2012*), ethylene (*Pan, Wang & Tian, 2002*), carbon monoxide (CO; *Xuan et al., 2008*), nitric oxide (NO), hydrogen peroxide ($H_2O_2$), $Ca^{2+}$ ions, calmodulin (*Liao et al., 2012*), hydrogen sulfide ($H_2S$; *Lin et al., 2012*), hydrogen ($H_2$; *Lin et al., 2014*) and methane (*Cui et al., 2015a*; *Cui et al., 2015b*) have been associated with the initiation and development of adventitious roots. More studies about signal transduction will help us understand the mechanisms involved in adventitious rooting.

Despite its simple two-carbon structure, ethylene (ETH) plays an important role in regulating plant growth and development. ETH may induce diverse effects throughout the plant life cycle, including seed germination, flower senescence and fruit ripening (*Khan et al., 2017*). ETH is also involved in the plant's defense reactions against pathogens or elicitor attacks, and in its responses to abiotic stresses such as wounding, hypoxia, chilling, freezing and flooding (*Khan et al., 2017*), indicating that ETH might be a regulator of plant growth, development and stress responses.

Since it was firstly discovered that ethylene stimulates adventitious root formation, there have been a great number of studies on the topic. *Riov & Yang (1989)* investigated the role of ETH in adventitious root formation in auxin-induced cuttings of *Vigna radiata* (L.) and found that ETH increased the number of adventitious roots while aminoethoxyvinylglycine (AVG, a kind of ETH inhibitor) significantly inhibited rooting. *Pitts, Cernac & Estelle (1998)* used the mutations of ETH and auxin to prove that auxin and ETH are required for normal root hair elongation. *Clark et al. (1999)* found that aboveground adventitious roots were produced in few ETH-insensitive never-ripe tomato plants and ETH- insensitive transgenic petunia plants compared to wild type plants, and proved the positive role of ETH in adventitious root formation. Similar results were obtained from sunflower cuttings (*Liu, Mukhcrjee & Reid, 1990*), mung bean hypocotyl cuttings (*Pan, Wang & Tian, 2002*), Panax ginseng C. A. Meyer (*Bae et al., 2006*), and globe artichoke seedlings (*Shinohara, Martin & Leskovar, 2017*). However, some researchers have indicated that ETH can inhibit adventitious root formation. *Biondi et al. (1990)* showed that in *Prunus avium* shoot cultures, the addition of ethylene precursor1-aminocyclopropane-1-carboxylic acid (ACC) decreased the rooting percentage while AVG caused roots to form. Some years later, the studies of *Ma et al. (1998)* demonstrated that $AgNO_3$ (another kind of ETH inhibitor) and AVG increased the number of roots in apple shoots.

Hydrogen gas ($H_2$), the lightest gas, is a colorless, odorless, and tasteless diatomic gas in nature. *Ohsawa et al. (2007)* found that $H_2$ can be used as an effective antioxidant therapy in the partial brain of rat, and since then, the use of $H_2$ in the prevention and control
of various diseases has been a hot topic of research (*Hong, Chen & Zhang, 2010*). More recently, the physiological functions of $H_2$ in higher plants have also been studied. $H_2$ has often been shown to respond to stresses like salt stress, drought stress, heavy metal toxicity, oxidative stress and UV-A irradiation (*Jin et al., 2013*; *Zeng, Ye & Su, 2014*; *Su et al., 2014*), and delaying fruit senescence (*Hu et al., 2014*) and inducing stomatal closure (*Xie et al., 2014*). Recently, another function of $H_2$ was discovered: the promotion of adventitious root occurrence. The results of *Lin et al. (2014)* demonstrated that 50% HRW could mimic hemin function and restore cucumber adventitious root formation by mediating the expression of auxin signaling-related and adventitious root formation-related genes. *Zhu et al. (2016)* has reported that HRW treatment increases the content of nitric oxide (NO) and up-regulated cell cycle activation during adventitious root formation. Moreover, *Cao et al. (2017)* suggested that HRW is involved in regulating auxin-induced lateral root formation by modulating NR-dependent NO synthesis.

Based on the results of the studies summarized above, ETH and $H_2$ are involved in adventitious root formation. However, the precise mechanism remains unclear. In this study, we hypothesized the exists of a relationship between ETH and $H_2$ during adventitious rooting in cucumber (*Cucumis sativus* L.).

To prove this hypothesis, ETH inhibitors were used. Proteomic methods have been widely applied to determine genetic and cellular function at the protein level. Many plant proteomic studies using 2-DE have found differential proteins during somatic embryogenesis, seed germination, adventitious root formation, flowering transition, and stem dormancy. Proteomic analysis has also proven useful in the identification of proteins associated with a range of biotic and abiotic stress responses (*Cui et al., 2015a*; *Cui et al., 2015b*; *Xu et al., 2016*). Therefore, 2-DE and matrix-assisted laser desorption/ionization time-of-flight/time-of-flight (MALDI-TOFTOF) were used to compare quantitative and qualitative changes that occurred during ethylene-$H_2$-induced adventitious rooting in the proteome. Through the comprehensive analysis of the function and content of differential proteins under different treatments and the expression of genes related to these proteins, we speculate that ETH possibly affect the expression of some proteins and the expression of their related genes. The results provide new insights into the physiological and molecular mechanisms associated with adventitious root development.

## MATERIAL AND METHODS

### Plant materials and treatments

Cucumber (*Cucumis sativus* L. var. 'Xinchun 4') seeds were purchased from Gansu Agricultural Institute (Lanzhou, China). The seeds were surface-sterilized in 5% sodium hypochlorite for 10 min and washed with water. The seeds were germinated on filter paper with distilled water in Petri dishes (15 cm-diameter, 2.5 cm-deep) and maintained at $25 \pm 1°$ C for 7 d with a 14-h photoperiod (photosynthetically active radiation $= 200\ \mu$mol m$^{-2}$s$^{-1}$). The 7-day-old seedlings with primary roots removed were used as explants and were maintained under the same temperature and photoperiod conditions for another 7 days in the presence of different media as indicated below. We described these different

treatments as (I) HRW (0, 1%, 10%, 50% and 100%); (II) ethephon (0, 0. 1, 0. 5, 1, 10 and 50 μmol L$^{-1}$); (III) AVG (aminoethoxyvinylglycine, 0, 0. 1, 1, 5 and 10 μmol L$^{-1}$); (IV) AgNO$_3$ and NaNO$_3$ (0, 0. 1, 0. 3 and 0. 5 μmol L$^{-1}$); and (V) 50% HRW + 0. 5 μmol L$^{-1}$ ethephon, 1 μmol L$^{-1}$AVG, or 0. 1 μmol L$^{-1}$ AgNO$_3$. After applying treatment explants were sampled to determine root number per explants. The hypocotyls (5-mm-long segments of the hypocotyl base where the adventitious root develops) of the explants after 48 h of treatment were immediately used or frozen in liquid nitrogen for further analysis.

## Preparation of hydrogen-rich water (HRW)

Purified H$_2$ gas (99. 99%, v/v) generated from a hydrogen gas generator (QL-300, Saikesaisi Hydrogen Energy Co., Ltd., China) was bubbled into distilled water at a rate of 300 mL min$^{-1}$ for 30 min. The corresponding HRW was rapidly diluted to the required saturations (*Zhu et al., 2016*). The H$_2$ concentration analyzed by gas chromatography in freshly prepared HRW was 0. 68 mM, which was defined as 100% HRW, and was maintained at a relatively constant level in 25 °C for at least 12 h.

## Quantification of ethylene

ETH was detected by a gas chromatographic analyzer (SP-3420A, Beifenruili Inc., China). Five fresh cucumber explants were placed in 20 mL vials, stoppered with secure rubber caps, and containing 1 mL culture solution. These vials had the same conditions as before when the seedlings grew for 12 h. A gas sample (2.5 mL) was extracted and injected into a gas chromatograph, which was fitted with a GDX-502 column (2 m × 3.2 mm) and a flame ionization detector. The temperature of the column, inlet and detector were 50, 140 and 240 °C, respectively. The flow rate of nitrogen gas, the carrier gas, was 30 mL min$^{-1}$. ETH production was expressed as μL h$^{-1}$ kg$^{-1}$FW.

## Protein extraction

The trichloroacetic acid (TCA)/acetone method was used to extract the total protein (*Damerval et al., 1986*). Approximately 2.0 g of the sample was manually ground to a fine powder in liquid nitrogen with 0.04 g crosslinking polyvingypyrrolidone (PVPP) and split into 6 tubes. The powder was suspended in 2 mL of acetone containing 10% w/v TCA and 0. 07% v/v β-mercaptoethanol (β-ME). After precipitation at −20 ° C for at least 8 h or overnight, the pellet was collected by centrifuging at 12,000× g for 30 min. Then the powder was suspended in two mL of ice cold acetone (−20 °C) containing 0. 07% (v/v) β-ME at 4 °C for 1 h, then centrifuged at 12,000× g for 20 min, and washed two times. The powder was then suspended in two mL of 80% acetone containing 0. 07% (v/v) β-ME at 4 °C for 30 min, centrifuged at 12,000× g for 15 min, and washed two times. The supernatant was removed and the pellets were dried in a freezer dryer. After air-drying, the powder was dissolved in a hydration solution [7 M urea, 2 M thiourea, 4% (w/v) 3-[(3-Cholamidopropyl)dimethylammonio]propanesulfonate (CHAPS), 1% (w/v) DL-Dithiothreitol (DTT)] at a normal temperature for 2 h, then centrifuged at 12,000×g for 30 min. The supernatant was the total protein. The protein content was determined calorimetrically using bovine serum albumin as a standard. Protein samples were stored at −80 °C for further analysis.

## 2-DE and staining

Sample aliquots containing protein were applied to 17 cm pH 4–7 immobilized pH gradient (IPG) strips, and small volumes of lysis buffer were added to the sample aliquots. Isoelectric focusing was performed on a PROTEAN IEF Cell system (Bio-Rad) for a total of 76 kVh at 20 °C. The voltage was set at 50 V for 14 h, 250 V for 1. 5 h, 1000 V for 2. 5 h, 9000 V for 5 h, and were run at 90000 V and then 500 V for a maximum of 24 h. Next, the gels were run using the PROTEAN II xi cell system (Bio-Rad) at 100 V for 1 h, followed by 250 V until the bromophenol blue nearly reached the bottom of the gel. After electrophoresis, staining was performed by placing the gels into Coomassie brilliant blue solution for 17 h. To provide support for the cucumber proteomics, we performed system optimization of the pH IPG strips, loading quantity and separation gel concentration. The protein spots on the pH 3–10 gels show that the acidic side (pH 3–5) in the region of the protein particle distribution was less than alkaline side (pH 9–10) area, forming a longitudinal tail (Fig. S1). Most of the explant protein spots were focused in the pH 4–7 region. The results showed that 800 μg of loading quantity was optimal (Fig. S2), and 12% (w/v) gel concentration separation was the best choice to obtain the good resolving effect (Fig. S3). The results also showed that pH 4–7 IPG strips (17 cm, nonlinear; Bio-Rad), 800 μg loading quantity and 12% (w/v) separation gel concentration was the most suitable combination for isoelectric focusing of the explants' total protein in cucumber.

## Image analysis and protein identification

The proteins in the gels were visualized by CBB-G250 staining. The gel images were obtained at a resolution of 300 dpi and then imported into PDQuest 8. 0 (Bio-Rad). The protein spots were excised from CBB-stained preparative polyacrylamide gels. The MS and MS/MS data for protein identification were obtained with a MALDI-TOFTOF instrument (*Zhang, Zhang & Zhou, 2015*). The database NCBI (https://www.ncbi.nlm.nih.gov/) were used to match and identified the protein spots. According to Gene Ontology and UniProt Protein Knowledgebase (http://www.uniprot.org/), the gene ontology (GO) analysis was performed on the proteins identified by mass spectrometry. The proteomic data have been uploaded to the Supplemental Files.

## Total RNA extraction and real time PCR

Total RNA from each sample was isolated using a MiniBEST Plant RNA Extraction Kit (TaKaRa, Beijing, China). Total RNA was reverse-transcribed by using the PrimeScript[TM]RT Master Mix (Perfect Real Time) according to the manufacturer's manual (Takara, Beijing, China). Quantitative real-time PCR was performed with a LightCycler 96 Real- Time PCR System (Roche, Switzerland) using SYBR *Premix Ex* Taq[TM] II (TaKaRa, Beijing, China). Every PCR reaction was performed twice, with three independent replicates. The relative expression level of each gene was acquired using a comparative Ct method followed by internal control normalization. PCR was carried out in 20 μL volumes using the following amplification protocol: 10 s at 95 °C, followed by 40 cycles of 15 s at 95 °C, and then annealing at 60 °C for 30 s. The relative quantization of gene expression was calculated and normalized to Actin. The expression change was calculated using the $2^{-\Delta\Delta ct}$ method. The primers used for PCR analysis are listed in Table 1.
**Table 1** List of the genes whose transcription profile was evaluated by RT-PCR.

| Gene name | NCBI accession number | primer | 5′–3′ primer sequence |
|---|---|---|---|
| CsACS3 | AB006805.1 | F | 5′-CCTTGCAGAGGCTGGCGATG-3′ |
| | | R | 5′-GGTGACTTGGAAGCCGTTGGAG-3′ |
| CsACO1 | AB006806.1 | F | 5′-AGGTAGGTGGCCTGCAACTCC-3′ |
| | | R | 5′-CTCCGAGGTTGACGACAATGGC-3′ |
| Rubisco | NP_198657.1 | F | 5′-GAGATTGAGGAGGCTAAGAAGGAA-3′ |
| | | R | 5′-GGGCTTGTAGGCGATGAAAC -3′ |
| SBPase | NP_001267658.1 | F | 5′-GAGTTCGTTATTTGGGGAGTCATT -3′ |
| | | R | 5′-TTATCAGGGGTTGCTTTGGTG -3′ |
| TDH | XM_004143321.2 | F | 5′-CCATTCAACTTTCCAACAGAACC -3′ |
| | | R | 5′- CGAGCCATCAACAACAGCA-3′ |
| CAPX | NM_001280706 | F | 5′-TTGGCTGGTGTTGTTGCTGT -3′ |
| | | R | 5′-GGCTCGGGTTTGTCCTCTCT -3′ |
| PDI | XM_004149750.2 | F | 5′- TGAGTTTTACGCCCCTTGGT-3′ |
| | | R | 5′-TCTCTGTTTGACTCCTCGTTGG -3′ |
| OEE1 | XM-004141898.2 | F | 5′-TTTGAAGTTGGTGCTGATGGTT-3′ |
| | | R | 5′- GGTGAAGAGGAACGGGACA-3 |
| Actin | DQ115883.1 | F | 5′-TGGACTCTGGTGATGGTGTTA-3′ |
| | | R | 5′-CAATGAGGGATGGCTGGAAAA-3′ |

# RESULTS

## Effects of different concentrations of HRW and ethephon on adventitious root development

To understand the effect of $H_2$ on adventitious root development, cucumber explants were treated with different concentrations of HRW (0, 1%, 10%, 50% and 100%). As shown in Fig. 1, different concentrations of HRW affected the development of adventitious roots. The cucumber explants treated with 50% and 100% HRW produced more adventitious roots than the control explants, but the root number under the 1% HRW treatment was significantly lower than that of the control. The maximum root number was observed with 50% HRW treatment (Fig. 1) and this was used for further study.

Different concentrations of ethephon, ETH-releasing compound, also had significant effects on the development of adventitious roots (Fig. 2). Lower concentrations of ethephon (0. 1, 0. 5, 1, and 10 $\mu$mol $L^{-1}$) increased adventitious root number, while the high concentration (50 $\mu$mol $L^{-1}$) significantly inhibited adventitious roots. The maximum inducible response was observed in 0. 5 $\mu$mol $L^{-1}$ ethephon-treated explants (Fig. 2). Thus, we used 0. 5 $\mu$mol $L^{-1}$ to further investigate the role of ethylene in rooting.

## Effects of ethylene inhibitors AVG and AgNO$_3$ on H$_2$-induced adventitious root development

To further study the function of ETH in adventitious root formation, ethylene inhibitors were used. Results shown in Figs. S4 and S5 indicate that various concentrations of ETH synthesis inhibitors AVG and AgNO$_3$ have negative effects on rooting. We used 1 $\mu$mol $L^{-1}$

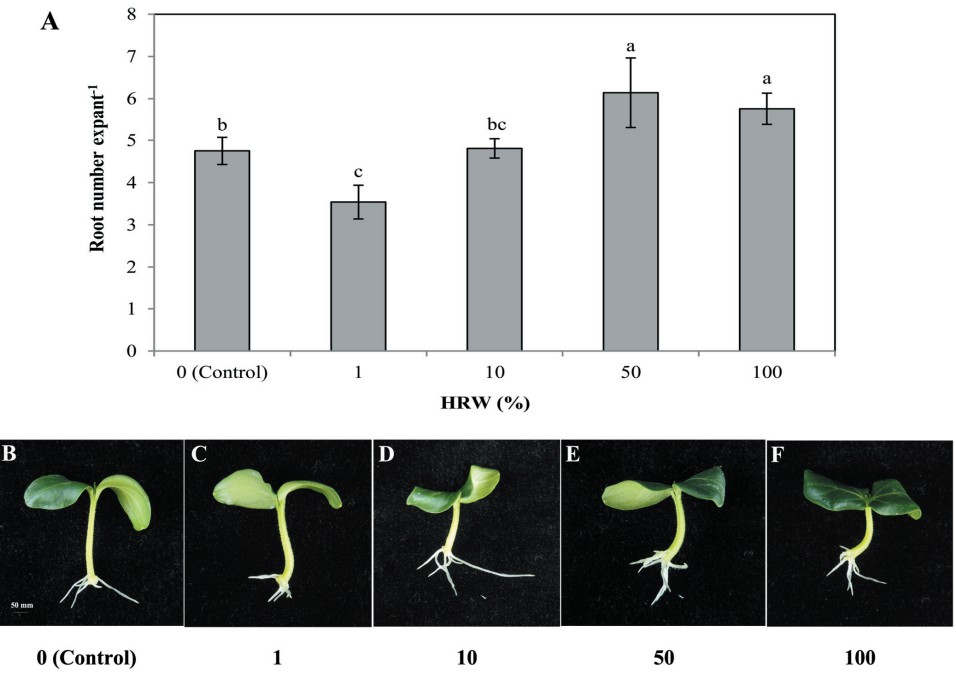

**Figure 1** **Effects of different concentrations of HRW on adventitious root development.** The primary root system was removed from hypocotyls of 7-day-old germinated cucumber. Explants were incubated with distilled water or different concentrations of HRW and indicated for 7 days. The number (A) of adventitious roots per explant were expressed as mean ± SE ($n = 20$ explants from each of four independent experiments). Bars with different lowercase letters were significantly different (Duncan's multiple range test, $P < 0.05$. Photographs (B–F) show hypocotyl explants after 7 days of the treatments indicated.

AVG and 0. 1 $\mu$mol L$^{-1}$AgNO$_3$ for further study. As seen in Fig. 3, explants treated with either HRW or ethephon alone showed improved adventitious rooting when compared with the control. Although explants treated with HRW plus ethephon had higher root number, they were not significantly higher than explants treated with either HRW or ethephon alone. When AVG (inhibitor of ETH synthesis) or AgNO$_3$ (an ETH action inhibitor) was added to the HRW, the inducing effects of ethephon on adventitious rooting were reversed (Fig. 3). Thus, we deduce that ETH might be involved in H$_2$-induced adventitious root development.

## Effect of H$_2$ on ETH production and ETH-related genes' expression

As shown in Fig. 4A, the production of ETH in 50% HRW treatment was about double that of the control. Moreover, the results in Fig. 4B indicate that HRW treatment significantly enhances the gene expression of *CsACS3*. When compared with the control, the expression of *CsACO1* was noticeably improved by HRW. These consistent results suggested that ETH might function in the downstream of H$_2$-induced adventitious root formation.

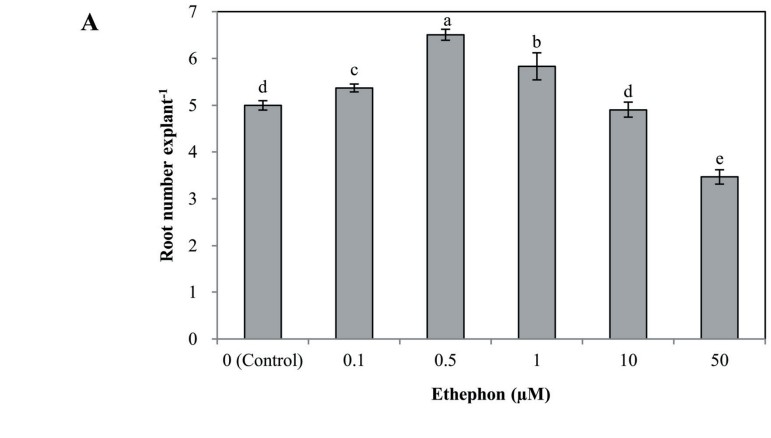

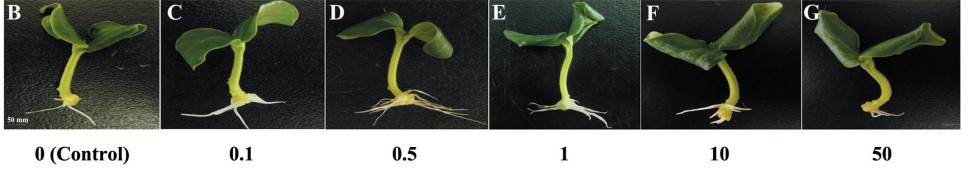

**Figure 2 Effects of different concentrations of ethephon on adventitious root development.** The primary root system was removed from hypocotyls of 7-day-old germinated cucumber. Explants were incubated with distilled water or different concentrations of ethephon and indicated for 7 days. The number (A) of adventitious roots per explant were expressed as mean $\pm$ SE ($n = 20$ explants from each of four independent experiments). Bars with different lowercase letters were significantly different (Duncan's multiple range test, $P < 0.05$. Photographs (B–G) show hypocotyl explants after 7 days of the treatments indicated.

## Comparison of 2-DE gels and protein identification during adventitious root development

All 70 protein spots from the 2-DE gels showed statistical differences. However, due to limited genomic information about *C. sativus* in the NCBI database, only 48 differentially expressed protein spots could be identified by MALDI TOF/TOF MS/MS with a significant search score, shown in Table 2 and Fig. 5. Out of the 48 differential proteins selected by analysis, 15 proteins were down-regulated while 9 (spot 203, 230, 858, 158, 975, 1116, 1132, 626 and 1140) were up-regulated in the HRW treatment; 10 proteins were down-regulated and 4 (spot 2129, 1908, 1561, 1284) were up-regulated in the ethephon treatment; 9 proteins were down-regulated whiles 1 (spot 239) was up-regulated in the HRW + AVG treatment. Table 2 shows a list of the putative names of proteins, matching species, protein score, accession number, molecular weights (MW), isoelectric point (PI), peptides count and Up/Down regulation. The results indicate that matching species listed above originated from the cucumber database, with molecular weights ranging from 9.4 to 75.3 KDa, PI of spots ranging from 4.31 to 8.6, protein scores ranging from 62 to 905, and matched peptide counts between 2 and 27.
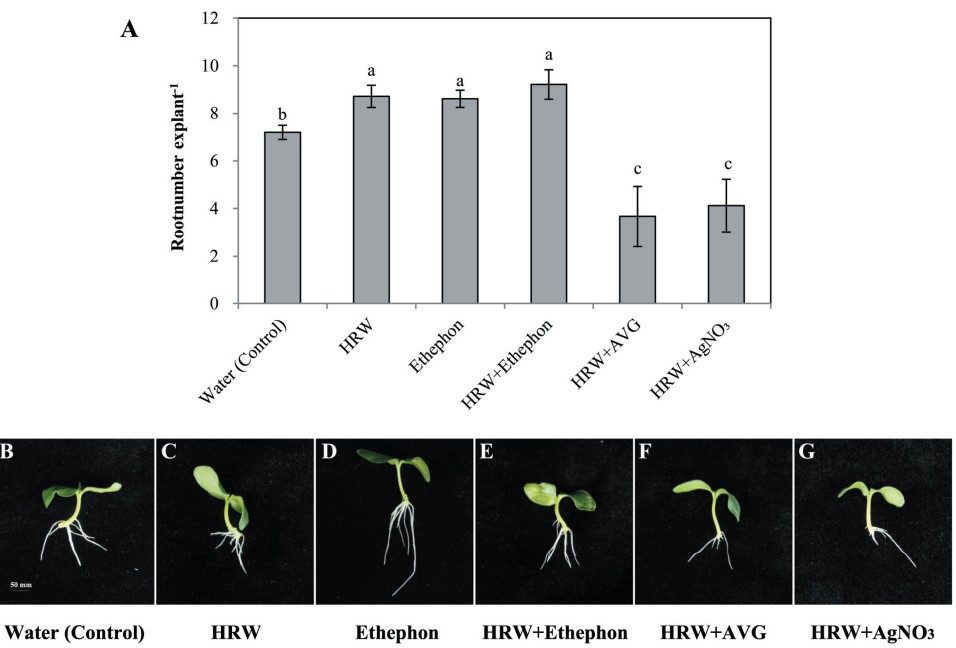

**Figure 3** **Effects of ETH inhibitor on $H_2$-induced adventitious root development.** The primary root system was removed from hypocotyls of 7-day-old germinated cucumber. Explants of cucumber were incubated with 0.5 μM ethephon, 50% HRW, 50% HRW + 0.5 μM ethephon, 50% HRW + 1 μM AVG, 50% HRW+ 0.1 μM AgNO₃/NaNO₃ and indicated for 7 days. The number (A) of adventitious roots per explant were expressed as mean ± SE ($n = 20$ explants from each of four independent experiments). Bars with different lowercase letters were significantly different (Duncan's multiple range test, $P < 0.05$. Photographs (B–G) show hypocotyl explants after 7 days of the treatments indicated.

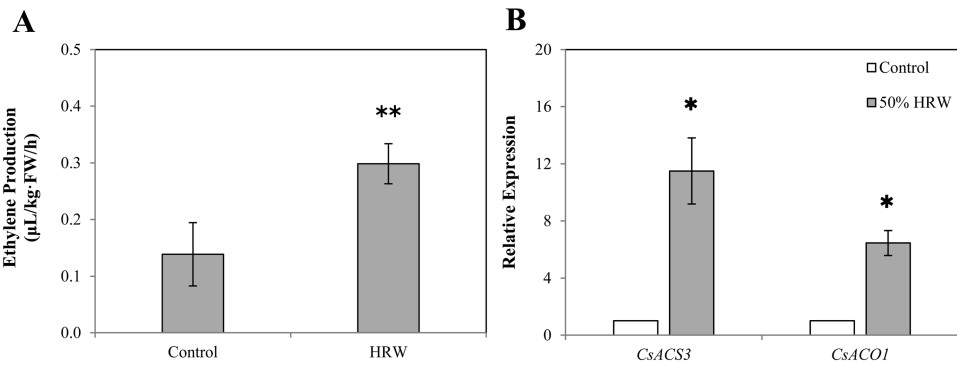

**Figure 4** **The effect of HRW on the ETH production (A) and on the expression level (B) of two ETH-related genes in cucumber explant.** The values (means ± SE) are the average of three independent experiments. Bars with an asterisk presents significant ($P < 0.05$), and two asterisks presents very significant ($P < 0.01$).

## Functional analysis of the six identified proteins

Forty-eight protein spots were identified using the annotated NCBI database, and these were designated as 41 known proteins and seven 7 unknown proteins. Using the Gene Ontology and UniProt Protein Knowledgebase, all of the identified proteins were categorized into eight 8 major groups (Fig. 6). Among these, Seventeen photosynthesis-related proteins constituted the largest percentage (35.4%), followed by six energy metabolism-related proteins (12.5%), six translation and transcription-related proteins (12.5%), six protein folding, modification and degradation-related proteins (12.5%), three stress response-related proteins (6.3%), two amino and metabolism-related proteins (4.2%), one cellular cytoskeleton-related proteins (2.1%), and seven unknown proteins (14.6%).

The differentially expressed proteins were identified using mass spectrometry and GO ontology analysis. These proteins were distributed in 3 categories: "biological process" (the largest group), "cellular component" and "molecular function" (the smallest group). As shown in Fig. 7 4 categories of proteins were identified in the biological process, comparing cellular process, metabolic process, response to stimulus and single-organism process; 3 categories of proteins were identified in molecular function, comparing: catalytic activity, transporter activity and binding; and 6 categories of proteins were identified in the cellular components, including organelles, cell parts and so on.

Out of the 41 proteins with known functions, 5 proteins were up-regulated in HRW or ethephon treatment, and were down-regulated in HRW + AVG treatment. These included ribulose-1,5-bisphosphate carboxylase small subunit (Rubisco), sedoheptulose-1,7-bisphosphatase (SBPase), threonine dehydratase (TDH), cytosolic ascorbate peroxidase (CAPX), and protein disulfide-isomerase (PDI) (Table 3). The oxygen-evolving enhancer protein (OEE1) was down-regulated under HRW or ethephon treatment and up-regulated under HRW + AVG treatment.

## Validation of the identified proteins at the mRNA level during adventitious root development

The mRNA levels of Rubisco, SBPase, TDH, CAPX and PDI genes under HRW or ethephon treatment were significantly higher than those of the control (Fig. 8). However, HRW+AVG treatment caused significant reduction in the expression of these genes. The mRNA levels of OEE1 increased significantly in the explants treated with HRW or ethephon, and decreased significantly in the explants treated with HRW+AVG. The changes of the mRNA levels of 6 genes were consistent with protein levels observed in the 2-DE results.

## DISCUSSION

Since *Renwick, Giumarro & Siegel (1964)* first reported that $H_2$ could promote germination, the function of $H_2$ in plants has been extensively investigated. Recently, $H_2$ has been linked to a wide variety of physiological processes in plants, ranging from the control of developmental processes to the regulation responses to abiotic stresses (*Xie et al., 2014*; *Hu et al., 2014*; *Wu et al., 2015*; *Chen et al., 2017a*; *Chen et al., 2017b*). *Lin et al. (2014)* and *Zhu & Liao (2017)* reported that $H_2$ plays positive roles in adventitious root development. In our study, suitable concentrations of exogenous $H_2$ significantly promoted the formation

Huang et al. (2020), *PeerJ*, DOI 10.7717/peerj.8896

**Table 2** Identification and analysis of proteins differentially expressed in response to ETH inhibitor on $H_2$-induced adventitious root development in cucumber explants.

| Spot No.[a] | Identified protein [b] | Species | Score [c] | NCBI accession number | MW [d] | Pi [e] | PeptidesCount [f] | Up/Down [g] |
|---|---|---|---|---|---|---|---|---|
| K9/1628 | Glutamine synthetase | *Cucumis sativus* | 406 | KGN57792.1 | 39251.8 | 5.82 | 13 | ↓ |
| K10/2129 | ribulose bisphosphate carboxylase/oxygenase precursor peptide | *Cucumis sativus* | 78 | AAA33131.1 | 21049.4 | 7.55 | 12 | ↑ |
| K11/1937 | Unknow protein | *Cucumis sativus* | 234 | KGN44928.1 | 22556.6 | 4.45 | 13 | ↓ |
| K12/1908 | Cytosolic ascorbate peroxidase | *Cucumis sativus* | 595 | KGN65655.1 | 27377.8 | 5.43 | 17 | ↑ |
| K13/1561 | Sedoheptulose-1,7-bisphosphatase | *Cucumis sativus* | 534 | KGN50632.1 | 42075.4 | 5.96 | 22 | ↑ |
| K14/1799 | Eukaryotic translation elongation factor 1 delta isoform X1 | *Cucumis sativus* | 417 | XP_011652060.1 | 24670.3 | 4.45 | 15 | ↓ |
| K15/1817 | Oxygen-evolving enhancer protein 1 | *Cucumis sativus* | 905 | KGN48464.1 | 34938.8 | 6.24 | 24 | ↓ |
| K16/2139 | PREDICTED:11S globulin seed storage protein 2-like | *Cucumis sativus* | 458 | XP_011652795.1 | 54382 | 5.71 | 11 | ↓ |
| K17/2062 | Cytochrome b6-f complex iron-sulfur subunit | *Cucumis sativus* | 112 | KGN43571.1 | 24255.2 | 8.45 | 9 | ↓ |
| K18/1169 | Chloroplast transketolase | *Cucumis sativus* | 712 | KGN56609.1 | 80569.6 | 6 | 27 | ↓ |
| K19/2156 | Ribulose bisphosphate carboxylase small chain | *Cucumis sativus* | 68 | KGN52085.1 | 20685.2 | 8.24 | 11 | ↓ |
| K20/1284 | ATPase alpha subunit (chloroplast) | *Cucumis sativus* | 385 | AAZ94637.1 | 55348 | 5.13 | 24 | ↑ |
| K21/2064 | ATPase alpha subunit (chloroplast) | *Cucumis sativus* | 246 | AAZ94637.1 | 55348 | 5.13 | 20 | ↓ |
| K22/361 | 26S protease regulatory subunit 6A homolog | *Cucumis sativus* | 230 | XP_004135596.1 | 38614.9 | 5.16 | 21 | ↓ |
| K23/748 | Oxygen-evolving enhancer protein 1 | *Cucumis sativus* | 752 | KGN48464.1 | 34938.8 | 6.24 | 20 | ↓ |
| K24/533 | Mg-protoporphyrin IX chelatase | *Cucumis sativus* | 236 | KGN49283.1 | 45728 | 5.72 | 20 | ↓ |
| L1/203 | Protein disulfide-isomerase | *Cucumis sativus* | 83 | KGN47715.1 | 57045.6 | 4.88 | 12 | ↑ |
| L2/230 | Threonine dehydratase | *Cucumis sativus* | 620 | KGN48214.1 | 67198.9 | 6.22 | 25 | ↑ |
| L3/1196 | PSII reaction center subunit V (chloroplast) | *Cucumis sativus* | 103 | AAZ94668.1 | 9380.7 | 4.83 | 4 | ↓ |
| L4/858 | Cytosolic ascorbate peroxidase | *Cucumis sativus* | 716 | KGN65655.1 | 27377.8 | 5.43 | 15 | ↑ |
| L6/656 | Proteasome subunit alpha type | *Cucumis sativus* | 466 | KGN62618.1 | 30861.4 | 4.98 | 9 | ↓ |
| L7/1009 | Unknow | *Cucumis sativus* | 412 | KGN57117.1 | 18646.5 | 4.56 | 9 | ↓ |
| L8/158 | Malic enzyme | *Cucumis sativus* | 496 | KGN66170.1 | 65093.8 | 5.72 | 23 | ↑ |
| L9/975 | Chlorophyll a-b binding protein, chloroplastic | *Cucumis sativus* | 270 | KGN51208.1 | 26517.6 | 5.84 | 7 | ↑ |
| L10/1116 | 60S acidic ribosomal protein P1 | *Cucumis sativus* | 355 | KGN56349.1 | 11414.7 | 4.34 | 3 | ↑ |
| L11/747 | Cysteine protease | *Cucumis sativus* | 114 | KGN62647.1 | 40614.9 | 5.2 | 8 | ↓ |
| L12/200 | Protein disulfide-isomerase | *Cucumis sativus* | 90 | KGN47715.1 | 57045.6 | 4.88 | 18 | ↓ |
| L13/527 | Mg-protoporphyrin IX chelatase | *Cucumis sativus* | 200 | KGN49283.1 | 45728 | 5.72 | 19 | ↓ |
| L14/1132 | Ribulose bisphosphate carboxylase small chain | *Cucumis sativus* | 100 | KGN52085.1 | 20685.2 | 8.24 | 11 | ↑ |
| L15/502 | Unknow | *Cucumis sativus* | 312 | KGN57401.1 | 31417.5 | 4.82 | 5 | ↓ |
| L16/626 | Sedoheptulose-1,7-bisphosphatase | *Cucumis sativus* | 546 | KGN50632.1 | 42075.4 | 5.96 | 18 | ↑ |
| L17/896 | PREDICTED: 29 kDa ribonucleoprotein, chloroplastic | *Cucumis sativus* | 298 | XP_004137828.1 | 30479.5 | 5.84 | 7 | ↓ |
| L18/103 | Chloroplast HSP70 | *Cucumis sativus* | 412 | KGN61439.1 | 75350 | 5.18 | 25 | ↓ |
| L19/447 | 30S ribosomal protein S1, chloroplastic | *Cucumis sativus* | 496 | XP_004147619.1 | 45304.5 | 5.34 | 18 | ↓ |
| L20/1129 | PREDICTED: 60S acidic ribosomal protein P2-4 | *Cucumis sativus* | 93 | XP_004150488.1 | 11419.9 | 4.53 | 8 | ↓ |
| L21/982 | PREDICTED: 11S globulin subunit beta | *Cucumis sativus* | 479 | XP_011651441.1 | 57575.8 | 7.21 | 15 | ↓ |

Huang et al. (2020), *PeerJ*, DOI 10.7717/peerj.8896

**Table 2** (*continued*)

| Spot No.[a] | Identified protein [b] | Species | Score [c] | NCBI accession number | MW [d] | Pi [e] | PeptidesCount [f] | Up/Down [g] |
|---|---|---|---|---|---|---|---|---|
| L22/1140 | ribulose-1,5-bisphosphate carboxylase small subunit | *Cucumis sativus* | 113 | CCF55356.1 | 20713.2 | 8.24 | 15 | ↑ |
| L23/732 | PREDICTED: 14-3-3-like protein D | *Cucumis sativus* | 120 | XP_004136830.1 | 29379.5 | 4.76 | 10 | ↓ |
| M15/139 | PREDICTED: ruBisCO large subunit-binding protein subunit alpha | *Cucumis sativus* | 401 | XP_004145754.1 | 61399.9 | 5.06 | 26 | ↓ |
| M17/239 | Oxygen-evolving enhancer protein 1 | *Cucumis sativus* | 216 | KGN48464.1 | 34938.8 | 6.24 | 19 | |
| M18/1018 | PREDICTED: 60S acidic ribosomal protein P1 | *Cucumis sativus* | 170 | KGN56349.1 | 11414.7 | 4.34 | 2 | ↓ |
| M19/1030 | Cytosolic ascorbate peroxidase | *Cucumis sativus* | 407 | KGN65655.1 | 27377.8 | 5.43 | 17 | ↓ |
| M20/340 | ribulose-1,5-bisphosphate carboxylase small subunit | *Cucumis sativus* | 137 | CCF55356.1 | 20713.2. | 8.24 | 14 | ↓ |
| M21/162 | Threonine dehydratase | *Cucumis sativus* | 206 | KGN48214.1 | 67198.9 | 6.22 | 22 | ↓ |
| M23/462 | Sedoheptulose-1,7-bisphosphatase | *Cucumis sativus* | 250 | KGN50632.1 | 42075.4 | 5.96 | 18 | ↓ |
| M24/354 | Beta-form rubisco activase | *Cucumis sativus* | 241 | KGN50568.1 | 48292.6 | 8.19 | 19 | ↓ |
| N1/351 | Unknow | *Cucumis sativus* | 238 | KGN59544.1 | 36680 | 4.7 | 23 | ↓ |
| N2/117 | Protein disulfide-isomerase | *Cucumis sativus* | 62 | KGN47715.1 | 57045.6 | 4.88 | 16 | ↓ |

**Notes.**

[a] Spot numbers are arbitrary as assigned by the software of PD-Quest 8.0.

[b] The name and functional categories of the proteins using MALDI TOF-TOF MS.

[c] The Mascot score obtained after searching against the NCBI nr database.

[d,e] Theoretical molecular mass (MV) and isoelectric point (PI) of the identified protein.

[f] Number of peptides sequenced.

[g] " ↑" represents up-regulated expression, " ↓" represents down-regulated expression.

Spots K9/1628 to K21/2064 were the proteins in ETH treatment; K22/361 to L23/732 were the proteins in ETH treatment and the control; M15/139 to N2/117 were the proteins in HRW+AVG treatment

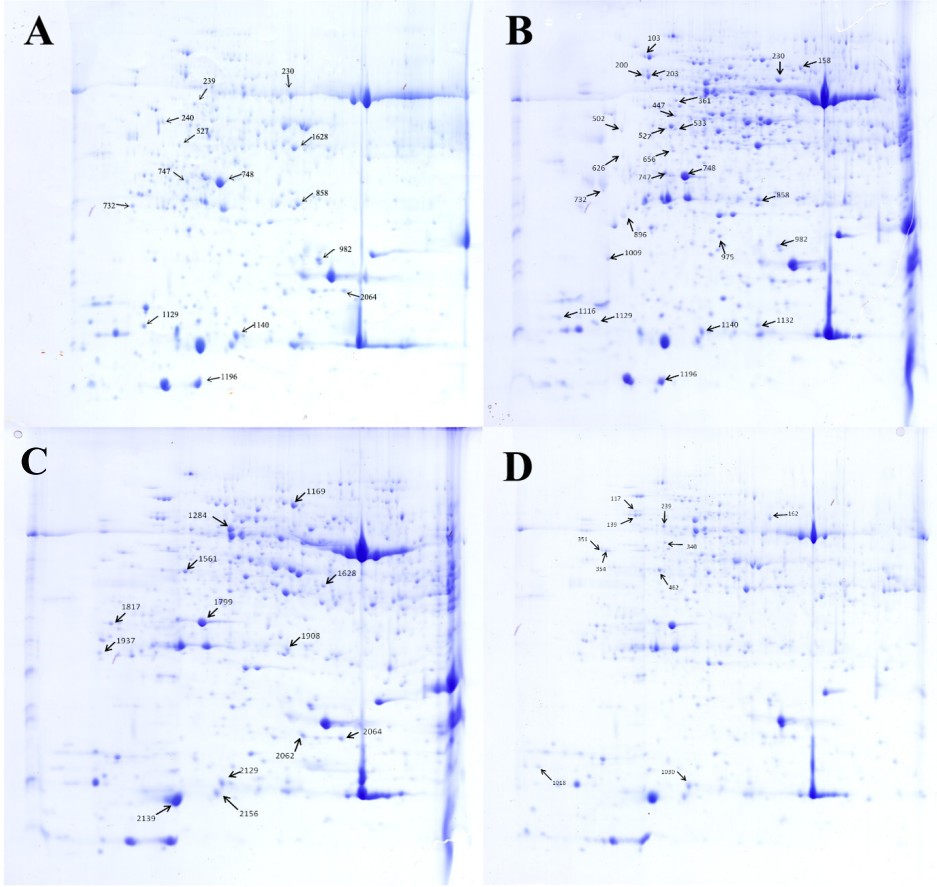

**Figure 5   Separation of the total proteins extracted from the explants of cucumber on 2-DE gels over the pH range 4–7.** Each map depicts one representative gel (of three replicates). A total of 48 protein spots showing difference are numbered on the gels. Arrows indicate the protein spots that were positively identified. (A) Control. (B) HRW. (C) Ethephon. (D) HRW + AVG.

of adventitious roots in cucumber. Moreover, ETH was also found to be involved in adventitious rooting. Previous studies showed that ETH stimulated adventitious root formation in some crops including mung bean (*Pan, Wang & Tian, 2002*), sunflower (*Liu, Mukhcrjee & Reid, 1990*) and marigold (*Jin et al., 2017*). *Xu et al. (2017)* revealed that ETH increased the formation of adventitious root in dose-dependent experiments. Our results suggested that exogenous ETH and 0. 5 $\mu$mol L$^{-1}$ ethephon significantly enhanced adventitious rooting. Evidently, both H$_2$ and ETH can induce adventitious rooting, but the interaction between H$_2$ and ETH during that process remains poorly understood. The results in this study suggest that ETH is involved in the H$_2$-induced adventitious rooting process and might function at the downstream of H$_2$.

We found that the selected ethylene inhibitors AVG and AgNO$_3$, caused partial inhibition of H$_2$-induced adventitious root development. When comparing HRW and the control in the production of ETH and the expression of two ETH-related genes, the HRW treatment
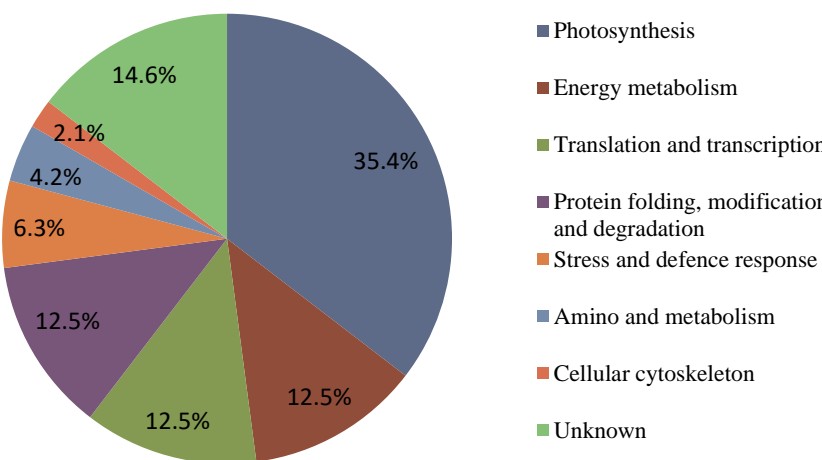

**Figure 6  The functional classification and distribution of all 41 identified proteins.** Unknown proteins include those whose functions have not been described.

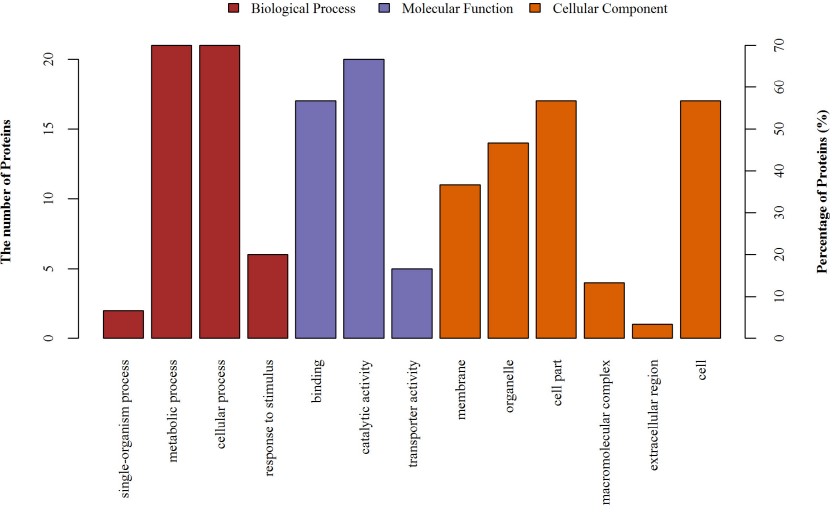

**Figure 7  The gene ontology (GO) analysis of the identified proteins.** Information of number and percentage of involved proteins in a term are shown in left and right y-axis.

was obviously higher than the control. These results suggest that ETH may be a downstream signal molecule in $H_2$-induced adventitious root development.

To further prove that ETH is involved in $H_2$-induced adventitious root formation, we used proteomic analysis, which allowed us to identify proteins whose expression was altered during adventitious rooting particularly in regard to $H_2$ and ETH treatments. The results of proteomic analysis showed that there were 41 differential proteins with known functions successfully identified in the adventitious root formation of cucumber. They were distributed among 3 categories. We also found 6 proteins, Rubisco, SBPase, TDH, CAPX, PDI and OEE1, to be used for further study.

**Table 3   Analysis of differential protein expression during H$_2$, ETH and AVG-induced adventitious root development.**

| Protein name | Functional category | Control | H$_2$ | ETH | H$_2$+AVG |
|---|---|---|---|---|---|
| ribulose-1,5-bisphosphate carboxylase small subunit | Photosynthesis | – | ↑ | ↑ | ↓ |
| Sedoheptulose-1,7-bisphosphatase | photosynthesis | – | ↑ | ↑ | ↓ |
| Threonine dehydratase | amino and metabolism | – | ↑ | – | ↓ |
| Cytosolic ascorbate peroxidase | stress response | – | ↑ | ↑ | ↓ |
| Protein disulfide-isomerase | Protein folding, modification and degradation | – | ↑ | – | ↓ |
| Oxygen-evolving enhancer protein 1 | photosynthesis | – | ↑ | ↑ | ↓ |

**Notes.**
↑represents up-regulated expression.
↓represents down-regulated expression.

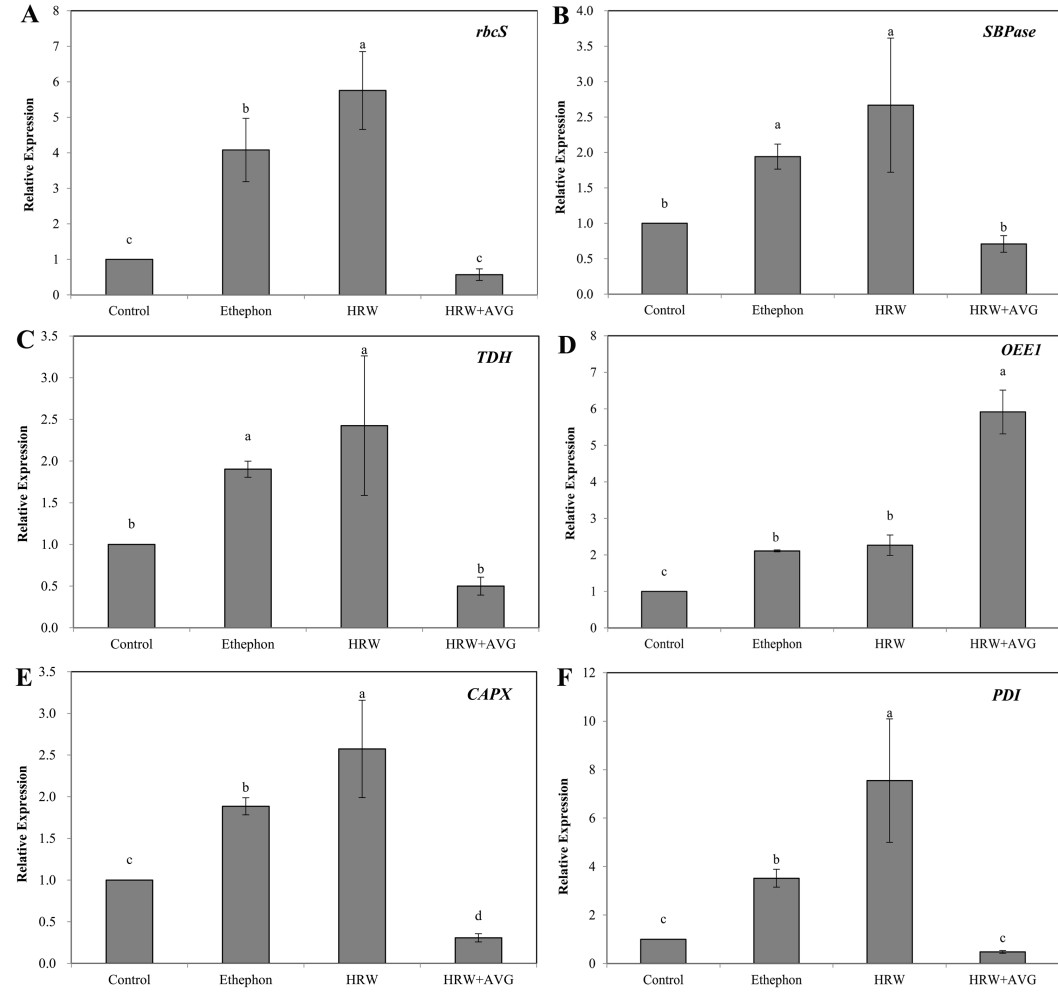

**Figure 8   The effect of ETH and HRW on the expression levels of *rbcS* (A), *SBPase* (B), *TDH* (C), *OEE1* (D), *CAPX* (E) and *PDI* (F).**  The values (means ± SE) are the average of three independent experiments. Bars with different lowercase letters were significantly different by Duncan's multiple range test ($P <$ 0.05).

It is known that Rubisco, SBPase and OEE1 proteins are related to the photosynthetic process. Rubisco is a vital enzyme associated with carbon fixation (*Xu, Gifford & Chow, 1994*) and its catalytic inefficiencies often limit plant productivity (*Andersson, 2008*; *Gunn, Valegard & Andersson, 2017*). SBPase has been shown to increase photosynthetic capacity and plant yield and it also promotes the growth of plants at the early growth stage (*Ogawa et al., 2015*). Our results show that Rubisco and SBPase expression were up-regulated under HRW or ethephon treatment, suggesting that $H_2$ and ETH may increase the efficiency of photosynthesis in explants by enhancing Rubisco and SBPase expression during rooting. ETH inhibitor AVG inhibited the inducing effects of HRW on Rubisco and SBPase, further indicating that ETH may play important roles in the regulation of $H_2$ during $H_2$-induced adventitious root formation.

OEE1 has been suggested to be involved in photosynthesis and PS II activities (*Yabuta et al., 2008*). *Chen et al. (2017a)* and *Chen et al. (2017b)* reported that OEE1 protein is necessary for oxygen evolving activity and is vital to maintain the stability of photosystem II. *Sugihara et al. (2000)* demonstrated that the level of OEE1 increased under NaCl treatment, and similar results were obtained at the transcription level. *Fatehi et al. (2012)* proposed that OEE1 was up-regulated in response to salinity, although in their study, $H_2$ and ETH treatment down-regulated OEE1 expression. This was similar to the results of *Bai et al. (2017)* who reported that the protein levels of OEE1 in oat leaves decreased in response to salinity. The possible explanation for this discrepancy is that OEE1 may change under different transcription, translation or treatment.

Amino acid metabolism has been found to play an important role in plant protein synthesis, photosynthesis, and development (*He et al., 2013*). TDH, a key enzyme in amino acid metabolism, contribute to both carbon skeleton supply and ammonium assimilation in plants and also plays an important role in metabolic acclimation (*Möckel, Eggeling & Sahm, 1992*). Our study suggests that $H_2$ up regulated TDH expression and ETH inhibitor AVG suppress the inducing effects. Proteomic analysis showed higher levels of TDH accumulation in wild ginseng than in cultivated ginseng (*Sun et al., 2016*). Moreover, proteomic analysis of rice and wheat coleoptiles revealed the potential role of amino acid biosynthesis in cellular anoxia tolerance (*Shingakiwells et al., 2011*). Thus, $H_2$ may increase carbon skeleton supply and metabolic acclimation in explants by enhancing TDH expression, which promotes adventitious root development. ETH is essential during this process. CAPX plays an essential role in the antioxidant defense mechanism (*Wu et al., 2014*) regulates plant growth, and induces organ formation (*Zhang et al., 2013*). The involvement of APX in fruit ripening and senescence has been proposed (*Torres et al., 2003*; *Giribaldi et al., 2007*), and *Caverzan et al. (2012)* demonstrated that APX genes can regulate both plant response to stresses and plant development. Additionally, *APX* mutant plant genes showed alterations in growth, physiology and antioxidant metabolism, suggesting the function of APX in plant development. Our results also show that CAPX expression was up-regulated by HRW or ethephon treatment, suggesting that $H_2$ may increase the reduction of oxidative damage and increase the productivity of explants by enhancing CAPX expression. However, ETH inhibitor AVG depressed the promotive effects of HRW on CAPX. PDI, related to protein folding, modification and degradation, is a multifunctional

protein in cells that assists many protein maturation processes (*Wilkinson & Gilbert, 2004*). It may catalyze protein thiol oxidation and restore and catalyze transformation disulfide bonds (*Houston et al., 2005*). PDI is also mainly associated with the protein secretory pathway in plants (*Irsigler et al., 2007*), and has been found to be down-regulated in rice under freezing stress (*Hashimoto & Komatsu, 2007*) and in soybean leaves under drought stress (*Irsigler et al., 2007*). In our study, PDI was up-regulated under HRW treatment and down-regulated under HRW+AVG treatment, suggesting that $H_2$ may increase the capacity of disulfide-bonded proteins by enhancing PDI expression. Meanwhile, ETH inhibitor AVG reverses the inducing effects of HRW on PDI, indicating that ETH may play important roles in H2-induced CAPX and PDI expression during rooting.

## CONCLUSION

Our results show that both $H_2$ and ETH play crucial roles in promoting adventitious root development. ETH might work as a downstream signaling molecule during $H_2$-induced rooting. Further proteomic studies show that photosynthesis-related proteins (Rubisco and SBPase), amino and metabolism-related protein (TDH), stress response-related protein (CAPX), and folding, modification and degradation-related protein (PDI) may play positive roles in the ETH-$H_2$-induced adventitious rooting process. Oxygen-evolving enhancer protein (OEE1) may play inhibiting roles during that process. The mechanisms underlying adventitious root development are very complex and more research will need to be carried out to fully understand adventitious rooting signaling.

## ACKNOWLEDGEMENTS

We thank Dr. Yongchao Zhu from Zhejiang University for providing guidance on experimental methods. We also thank Dr. Chen Bai from Gansu Academy of Agricultural Sciences for supplying a gas chromatographic analyzer.

### Funding

This work was supported by the National Key Research and Development Program (2018YFD1000800), the National Natural Science Foundation of China (Nos. 31860568, 31560563 and 31160398), the Research Fund of Higher Education of Gansu, China (2018C-14), the Natural Science Foundation of Gansu Province, China (Nos. 1606RJZA073 and 1308RJZA179), Scientific Research Foundation for the Yong Graduate Supervisor of Gansu Agricultural University in Lanzhou P. R. China (GAU-QNDS-201709) and Feitian and Fuxi Excellent Talents in Gansu Agricultural University. The funders had no role in study design, data collection and analysis, decision to publish, or preparation of the manuscript.

### Grant Disclosures

The following grant information was disclosed by the authors:
The National Key Research and Development Program: 2018YFD1000800.

The National Natural Science Foundation of China: 31860568, 31560563, 31160398.
The Research Fund of Higher Education of Gansu, China: 2018C-14.
The Natural Science Foundation of Gansu Province, China: 1606RJZA073, 1308RJZA179.
Scientific Research Foundation for the Yong Graduate Supervisor of Gansu Agricultural University in Lanzhou P. R. China: GAU-QNDS-201709.
Feitian and Fuxi Excellent Talents in Gansu Agricultural University.

## Competing Interests

The authors declare there are no competing interests.

## Author Contributions

- Dengjing Huang performed the experiments, analyzed the data, authored or reviewed drafts of the paper, and approved the final draft.
- Biting Bian conceived and designed the experiments, performed the experiments, analyzed the data, prepared figures and/or tables, authored or reviewed drafts of the paper, and approved the final draft.
- Meiling Zhang performed the experiments, analyzed the data, prepared figures and/or tables, authored or reviewed drafts of the paper, and approved the final draft.
- Chunlei Wang conceived and designed the experiments, analyzed the data, prepared figures and/or tables, authored or reviewed drafts of the paper, and approved the final draft.
- Changxia Li performed the experiments, analyzed the data, prepared figures and/or tables, and approved the final draft.
- Weibiao Liao conceived and designed the experiments, authored or reviewed drafts of the paper, and approved the final draft.

## Data Availability

All the raw data are available in the Supplemental Files.

## Supplemental Information

Supplemental information for this article can be found online at http://dx.doi.org/10.7717/peerj.8896#supplemental-information.

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
