# Peer review of "The role and proteomic analysis of ethylene in hydrogen gas-induced adventitious rooting development in cucumber (Cucumis sativus L.) explants"

_PeerJ, doi:10.7717/peerj.8896_

## Round 0.1 · original submission · Major Revisions

Dear Dr Liao and coauthors,

The reviewers and me found interesting data in your work and we are enthusiastic about its prospects. However, the manuscript requires serious major revisions that require your attention.

Although not related to the science, the writing and formatting of the paper is not an adequate. As the peer review process demands free time form reviewers and editors, making sure the English is correct and that the paper conforms with the formatting required by the publisher (including clear and independent figures- I could not even read Fig 9) will facilitate the work of everybody involved. Please pay attention to this as I will not process the revised manuscript if these aspects are not satisfactory.

The introduction needs work. I suggest the authors to build the case for the work on stronger theory about the role of hydrogen and ethylene on root formation. There is also a need to call for more representative citations, including some historical ones:

Biondi, S., Diaz, T., Iglesias, I., Gamberini, G., & Bagni, N. (1990). Polyamines and ethylene in relation to adventitious root formation in Prunus avium shoot cultures. Physiologia plantarum, 78(3), 474-483.

Pitts, R.J., Cernac, A. and Estelle, M., 1998. Auxin and ethylene promote root hair elongation in Arabidopsis. The Plant Journal, 16(5), pp.553-560.

Clark, D.G., Gubrium, E.K., Barrett, J.E., Nell, T.A. and Klee, H.J., 1999. Root formation in ethylene-insensitive plants. Plant Physiology, 121(1), pp.53-60.

The introduction should include a falsifiable hypothesis at minimum and ideally a prediction that results from such hypothesis.

The reviewers note excessive speculations in the manuscript. I agree with their assessment as often correlation seems to be confused with causation. Please note when data suggest or is consistent with a mechanism and we speculate/postulate a new role. Details on methods and data interpretation are given by the expert reviewers and must be addressed should you choose to send a revised manuscript.

I suggest you rethink the first paragraph of the discussion as it does not highlight the novelty of the work and it appears to show your results as confirmatory even for the same species. I would also avoid introduction material in this paragraph (e.g. first sentence).

I hope you can positively respond to all the comments in a revised manuscript

Reviewer 1 ·

Basic reporting

Some places exist formatting, gramma and other mistakes. It needs an improvement, Please polish it.

Experimental design

methods: The details (manufacturer, city, country) of the Kit used in RNA isolation should be added. The instrument and reagent used in real time PCR experiments should be stated.
How identified proteins can be “functional classification and distribution” ? please add the related methods.

Validity of the findings

no comment

Additional comments

The manuscript entitled: “The involvement of ethylene in hydrogen gas-induced adventitious rooting development in cucumber (Cucumis sativus L.) explants and its proteomic analysis” stated an interesting work on the role of hydrogen gas and ethylene in cucumber adventitious rooting. However, there are some concerns regard to this work.

1. Some places exist formatting, gramma and other mistakes. It needs an improvement, Please polish it.
2. Introduction:
the role of ethylene on adventitious root formation is not well stated in the introduction section. Some relevant work are suggested been introduced. Meanwhile, studies with AVG as a inhibitor can be introduced.
The research hypothesis should be clearly described in the end of the introduction section.
3. Materials and methods:
The full name of some abbreviations should be added where it first appearance, for example, TCA, PVPP, CHAPS, IPG, etc.
The details (manufacturer, city, country) of the Kit used in RNA isolation should be added. The instrument and reagent used in real time PCR experiments should be stated.
How identified proteins can be “functional classification and distribution” ? please add the related methods.
The unit “litre” should be uniformed to “L” or “l”.
4. Results:
Line 231, the quotation mark “” should be match.
Line 232, Figs. 8, 9.
5. Discussion:
A deepgoing discussion should be considered. What is the mechanism of H2 regulating ETH signal? How can H2 regulating identified 6 proteins expression? They all are in a ETH-dependent manner?
Line 306, “Considerably more work will need to be done to determine that the change in OEE1 protein abundance is dependent on whether high hydrogen or hypoxia.” Authors stated that the negative effects of H2 on OEE1 maybe duo to the high hydrogen or hypoxia, but the title is refer to “hydrogen gas”, so the further results of the hypoxia effects or additional controls should be given. The stress response-related protein (CAPX) was identified in HRW response, it also suggested to investigate the effects of hypoxia. In addition, is CAPX plays a role in development? If so, please discuss.
6. Acknowledgments:
No information? Please check.
7. References:
Some references are not in the correct format. For example: Line 367, 268-271. Line 374: Synechococcus sp. PCC 7002, please pay attention to the italicized. Line 390, lack of doi. Line 421, (Tagetes erecta L.), pay attention to the italicized. Similar mistakes also fund in Line 440, 454, 490, 492, 493, 504, etc.
8. Table 1: The reference gene Actin should be added as well as its sequnces.
9. Table 2: the meaning of the symbols “↑”, “↓” should be added. The A, B, C, and D in footnote should be appeared in the table (boxhead), or “A”, “B”, “C”, and “D” is suggested to be deleted.
10. Table 3: added the meaning of “↑”and “↓”. Why there have blanks in the rank of ETH?
11. Figure 1: the quality of Fig. 1 should be improved, and the scale bar in the picture should be added. Some pictures in Fig. 1B seems not typically selected as corresponding to the data in Fig. 1A.
12. Figure 2: there are difference at the adventitious rooting phenomenon (especially in root length) in Fig. 1 and Fig. 2, thus it seems that HRW is beneficial for AR length and ETH helps increase the number of AR? What is the mechanism behind the phenomenon? Please added the scale bar.
13. Figure 3 and 4: i think the dose effect of AVG and AgNO3 on adventitious rooting are not the key results in this paper, also, it lock of novelty. It suggested that been transfered this two figures to supplemental materials.
14. Figure 5: delet “and NaNO3 “ in the legend. The HRW, Ethephon in Fig. 5A should be consistent to that of “H2” , “ETH”in Fig. 5B, as well as the whole MS. Please added the scale bar.
15. Figure 8: Three items correspond to two ordinates, and what is their correspondence? It suggested that add the number of proteins on the top of each column.
16. Figure 9: figures in Fig. 9 needs to be revised, as the characters are too small to read.

Reviewer 2 ·

Basic reporting

1) I suggest that the article still need an English editing.
2) The Introduction section is too simple to provide sufficient field background. The authors should tell more researches that has been made about the role of ETH and H2 during AR formation.

Experimental design

3) Line 114: Which part of the explants did the samples take or all the explants were used?

Validity of the findings

4) I think the current result cannot draw a conclusion that ETH is the downstream signaling molecule in adventitious rooting induced by H2 unless the author can tell us the variation of ETH related gene expression after H2 treatment.

Additional comments

5) Lines 253 – 277: I suggest that the author can adjust the order of the sentences to make it more logical.
6) The author can simplify the context sometimes.
a. Lines 286 – 290: I suggest to delete the sentence “In a proteomic study of sorghum leaves under salt stress, Rubisco activity was enhanced in response to salt stress (Ngara et al. 2012). Andersson et al. (2008) reported that Rubisco expression could influence plant growth and lead to complex changes. Previous study showed that Rubisco gene grown at high-CO2 concentration might result in a decline in Rubisco activity in pea (Xu et al. 1994)”
b. Lines 291 – 293: These two sentences are almost identical.
7) I suggest the author to turn all English numbers into Arabic numerals including abstract section, lines 105, 108, 209 – 213, 222 – 227, 233 – 237.
8) It is unclear to me that how could you divide the 48 proteins into 8 groups in lines 220 – 222, Fig 7. Which database did you use?
9) The Fig 8 display the result of GO annotation but the legend is COG.
10) Lines 232 – 236: The result is inconsistent with the presentation of Fig 8. There were 4,3,6 categories in biological progress, molecular function and cellular component in Fig 8, respectively, but the author said 9,4,7 categories in the context.
11) Table 2: I suggest to divide the proteins into 4 groups according to your treatments.
12) Table 2: The legend is unclear to me. What is A, B, C D? And I suggest to explain the meaning of score, MW, PI in the legend.
13) Abstract: “photosynthesis -related proteins” should be “photosynthesis-related proteins”
14) Line 66: “hydrogen sulfide (H2S; Lin et al. 2012) hydrogen (H2;Lin et al. 2014)” should be “hydrogen sulfide (H2S; Lin et al. 2012), hydrogen (H2;Lin et al. 2014)”.
15) Line 75: Using the English comma.
16) Line 231: ‘biological process” should be “biological process”
17) Lines 322 -325: APX should be CAPX.

---

## Round 0.2 · Minor Revisions

Thank you for carefully working through the comments and providing us with a much more improved manuscript. Your introduction is now clear and the paper can be accepted provided some very minor changes are incorporated.

Figure 8 does not have enough quality and errors have been pointed out by the two reviewers about quantities and descriptions. Please fix the figure and increase its quality.

One of the reviewers identified some numerical errors in one of your paragraphs. Please fix so it is in accordance with the data presented.

These are minor and hopefully quick fixes.

Reviewer 1 ·

Basic reporting

The English writing has improved a lot, but some mistakes and format problems still to be noticed.

Experimental design

no comment

Validity of the findings

no comment

Additional comments

Authors have carefully revised the manuscript. However, there still have some minor comments.
1. Please double-check the error bar in Fig. 8D. Figure 8 is not of high quality, so it is recommended to redraw.
2. The statement of statistical analysis in Fig. 8 is incorrect, becuase there is no asterisk in the figure. What do the different letters mean? ANOVA analysis was used? How many parallel tests in each independent experiments? It should be included in all figure legends and Materials and Methods.
3. In the legend of Fig. 7, The number of involved proteins and the information of percentage in a term are shown in left and right y-axis respectively?
4. In Fig. 5, C should be in the suitable position.
5. Please double-check the numbers in Line 267-275. What is the mean of 36.935.4%? 4.34.2% 2.22.1% There are three 1312.5?
6. Please notice to the "et al." in Line 54, 73, 77, 79......
7. Please double-check the references. For example, in Line 499, "J Bacteriol, 174" ? in Line 401, "J Exp Bot, 59" ? Line 417, "78" do not be in italics. Line 429, "photosystem II manganese-stabilizing"
8. In Line 325, "H2" should be in the right format.
9. In Line 330, Rubisco, SBPase, TDH, CAPX, PDI and OEE1, total 6 proteins?

Reviewer 2 ·

Basic reporting

no comment

Experimental design

no comment

Validity of the findings

no comment

Additional comments

1. Line 30: “during”
2. Line 62: “stress”
3. Line 236: “Although”
4. Line 268: “All”
5. Line 269 – Line 274: Firstly, the percentage of all groups were wrong. For example, “Seventeen (17) photosynthesis-related proteins constituted the largest percentage (36.935.4%)”. 17/48 = 35.417%. And so on. Secondly, existing some formatting mistake, such as 36.935.4%, 1312.5%, 6.56.3%, 4.34.2% and so on. Thirdly, Line 274: there were 7 unknown proteins instead of 4. In addition, “six 6 energy metabolism-related proteins” should delete “six”. Please carefully check this section.
6. Fig 8: the percentage of unknown protein is 14.6% instead of 8.3%.

---

## Round 0.3 · Minor Revisions

Dr Liao, I am sorry about revisiting this. After editorial review the Section Editor for this part of the journal has commented and said:

"There are no links between the classifications and annotations used in Figures 6 and 7; they should have accompanying information linked to the sequence data being described. It should be emphasized that the NCBI annotations are matches and not original depositions, or are they? Otherwise sequence data relating to the measurements should be made available somewhere (perhaps in additional supplemental files). The manuscript should not be accepted until clear ties between classification and annotation can be tied to actual sequence data. i.e. I believe that more revisions are required."

I hope you can quickly address these final comments.

---

## Round 0.4 · accepted · Accept

Thanks for your quick turnaround. The editorial board is now happy that your annotations in supplemental materials are sufficient.